The effects of plasma from patients with active thyroid-associated orbitopathy on the survival and inflammation of melanoma-associated fibroblasts

Chen Huifang 1 2
Chen Shiyuan 2
Liu Zhenfeng 2 lzf18077229888@126.com
1 Department of Medicine, Guangxi University of Science and Technology , Liuzhou , China
2 Department of Traditional Chinese Medicine, The Second Affiliated Hospital of Guangxi University of Science and Technology , Liuzhou , China
Guan Fanglin
Electronic publication date: 2024 Nov 27
Publication date: 2024
Volume: 12
Electronic Location ID: e18612
Received 2024 Aug 26; Accepted 2024 Nov 8
Copyright: © 2024 Chen et al.
Copyright year: 2024
Copyright holder: Chen et al.
License: This is an open access article distributed under the terms of the Creative Commons Attribution License, which permits unrestricted use, distribution, reproduction and adaptation in any medium and for any purpose provided that it is properly attributed. For attribution, the original author(s), title, publication source (PeerJ) and either DOI or URL of the article must be cited.
License URL: https://creativecommons.org/licenses/by/4.0/

Keywords: Melanoma, Tumor microenvironment, Melanoma-associated fibroblasts, Inflammation, Survival

Funding: Guangxi Zhuang Autonomous Region Administration of Traditional Chinese Medicine GXZYB20220329 This project was supported by Guangxi Zhuang Autonomous Region Administration of Traditional Chinese Medicine (GXZYB20220329). The funders had no role in study design, data collection and analysis, decision to publish, or preparation of the manuscript.

==============================
Background

Plasma from patients with active thyroid-associated orbitopathy (TAO-A) could cause inflammation to fibroblasts, and such a mechanism was explored in the context of melanoma.

Methods

Plasma samples collected from TAO-A patients and healthy control (HC) were primarily co-cultured with the melanoma-associated fibroblasts (MAFs) derived from melanoma patients. The survival and inflammation of the co-cultured MAFs were measured after confirming the levels of pro-inflammatory cytokines. Ki67 and Vimentin (VIM) markers were analyzed by immunofluorescence, and cell survival and migration were assessed using cell counting kit-8 (CCK-8) and Transwell. The THP-1 cells were induced to differentiate into macrophages, which were subsequently co-cultured to assess M1/M2 polarization status. Meanwhile, the levels of inflammatory factor were detected by enzyme-linked immunosorbent assay (ELISA). The gene expression was measured by reverse transcription quantitative PCR (RT-qPCR), and the activation of PI3K/AKT, STAT1, p65, and ERK signaling pathways was detected by Western Blot.

Results

Plasmas derived from TAO-A patients were characterized by elevated levels of pro-inflammatory cytokines, which enhanced the inflammation status and survival of MAFs, promoted the levels of PI3K and AKT, and downregulated expression of Bax. The co-culture of the plasma with MAFs evidently promoted M1 polarization and the phosphorylation of STAT1, P65 and ERK1/2.

Conclusion

These findings proved the effects of the plasmas of TAO-A patients on the survival and inflammation of MAFs, providing evidence for future studies to delve into the relevant mechanisms.

Introduction

Recent studies have shown that the tumor microenvironment (TME) plays an indispensable role as a potential therapeutic target in cancers (Pitt et al., 2016), and that stromal cells of the TME are decisive to tumor progression and unsatisfactory treatment outcome (Junttila & de Sauvage, 2013). The TME contains both non-cancerous cells and components that could drive the growth, invasion, metastasis, and response of tumor cells to therapies (Xiao & Yu, 2021; Elhanani, Ben-Uri & Keren, 2023). A variety of cell types, including tumor-infiltrating immune cells, cancer-associated fibroblasts (CAFs) and endothelial cells of extracellular matrix, are present in the TME (Arneth, 2019; Meng et al., 2024; Zhang et al., 2024). Hence, a close interaction between the TME and tumor cells could promote tumorigenesis (Hajdara et al., 2023).

Thyroid-associated orbitopathy (TAO-A), also known as Graves’ ophthalmopathy, is an autoimmune disease correlated with thyroid dysfunction (Bahn, 2010). A distinguishing feature of TAO-A is the swelling of orbital tissue, which includes both extraocular adipose and muscle tissues. Soft tissue swelling is a result of the accumulation of nonsulfated glycosaminoglycans, inflammation, hyaluronan, and activation of local fibroblasts (Berchner-Pfannschmidt et al., 2016; Wang et al., 2023). If left untreated, the swelling of orbital tissue can lead to orbital congestion, exophthalmos, compressive neuropathy, and ultimately vision loss (Wang et al., 2021). Melanoma is a skin cancer resulted from malignant transformation of melanocytes, which are a type of cells with pigment-producing ability (Ahmed, Qadir & Ghafoor, 2020; Schadendorf et al., 2018). Patients at an early stage can be successfully treated with surgery but the survival outcomes could be sharply reduced by the occurrence of metastasis (Davis, Shalin & Tackett, 2019). Melanoma is one of the most immunogenic tumor types and is more likely to respond actively to immunotherapy due to its well-expressed lymphoid infiltration (Tagliaferri et al., 2022; Marzagalli, Ebelt & Manuel, 2019). However, similar to other cancer types, melanoma cells acquire a variety of suppressive properties to escape from both innate and adaptive immune recognition and destruction (Marzagalli, Ebelt & Manuel, 2019).

Melanoma-associated fibroblasts (MAFs) refer to a melanoma-driven distinct subpopulation of CAFs that promote tumorigenesis through facilitating immune evasion and proliferation of tumor cells (Papaccio et al., 2021; Eble & Niland, 2019). Though MAFs share some basic characteristics with CAFs, MAFs also undergo some molecular changes to adapt their functions to particular requirements of melanoma cells (Romano et al., 2021), significantly contributing to the structural alternations of microenvironment and some molecular and cellular alternations related to the outcome of melanoma (Bellei, Migliano & Picardo, 2020). MAFs enhance the proliferation and invasiveness of melanoma cells and promote their resistance to anti-apoptosis by secreting growth factors (e.g., TGF-β) and extracellular matrix proteins (Brenner et al., 2005). In addition, MAFs can secrete pro-angiogenic factors to increase the formation of neovascularization in the tumor, providing nutrients and oxygen to the tumor to support its growth and spread (Sala et al., 2002). At present, most studies have analyzed the tumor-promoting role of MAFs in melanoma (Lian et al., 2024; Shi et al., 2023). This study set out to explore the effects of plasma from TAO-A patients on the survival, migration and inflammatory response of MAFs, and to further analyze whether such an effect was mediated through modulating PI3K/AKT signaling pathway and promoting M1 polarization in macrophages. The current findings contributed to the understanding on the association between TAO-A and tumors and provided novel targets for future research and therapy for melanoma.

Materials and Methods

Ethic statement

The current study obtained the approval from the Ethic Committee of our hospital and all the participants enrolled in this study signed the written informed consent. All the experiments were carried out strictly following relevant regulations and guidelines. The study was approved by Guangxi University of Science and Technology Medical Ethics Committee (No. YX20230301H015).

Human sample collection

Human blood samples were collected from patients with TAO-A and HC (n = 5 for each group) as needed. In detail, 4 mL venous blood was collected into the blood collection tubes containing ethylenediaminetetraacetic acid (EDTA, E8040; Solarbio Lifesciences, Beijing, China). Then the plasma was obtained via centrifugation at 3,000 rpm for 10 minutes (min) at 4 °C and stored at −80 °C for subsequent analysis.

MAFs used in this assay were isolated from primary tumors of melanoma patients (n = 3) (Çakır et al., 2021). In detail, the inner tumor mass was minced into approximately 1 mm3 and digested in 20 mL Dulbecco’s modified Eagle’s medium (DMEM, 21063-029; Thermo Fisher Scientific, Waltham, MA, USA) supplemented with 0.6 U/mL dispase (17105-041; Thermo Fisher Scientific, Waltham, MA, USA) and 200 U/mL collagenase IV (17104-019; Thermo Fisher Scientific, Waltham, MA, USA). Differential adhesion/trypsinization method was employed to separate MAFs from the melanoma cells. Briefly, the tumor cell suspension digested in collagenase IV and dispase was plated into the plastic cell culture dishes for 30 min. Hereafter, adherent cells were cultured for differential adhesion after the removal of floating cells. After that, subconfluent cell culture was trypinsized for 1 min to eliminate detached cells, while the adherent cells enriched in MAFs were subcultured for differential adhesion again. For the assays in this study, MAFs grown until the confluence of 75–80% were rinsed in phosphate-buffered saline (PBS) and further cultured in 10 mL basal medium consisting of standard DMEM, 20% fetal bovine serum (FBS, 10437-036; Thermo Fisher Scientific, Waltham, MA, USA), 1% penicillin-streptomycin (15070-063; Thermo Fisher Scientific, Waltham, MA, USA) and 1% L-glutamine (21051-024; Thermo Fisher Scientific, Waltham, MA, USA).

Cell culture and induction

Here, we used THP-1 monocytes as a macrophage model to investigate the effect of plasma on macrophage polarization in TAO-A patients. The complete medium containing 90% Roswell Park Memorial Institute-1640 medium (11875-093; Thermo Fisher Scientific, Waltham, MA, USA), 10% FBS and 0.05 mM 2-mercaptoethanol (21985-023; Thermo Fisher Scientific, Waltham, MA, USA) was used to culture human monocytes THP-1 (SCSP-567; National Collection of Authenticated Cell Cultures, Shanghai, China) at 37 °C with 5% CO2. All the cells were identified by STR profiling and tested negative for mycoplasma contamination. For the induction into macrophages, THP-1 monocytes were treated with phorbol-12-myristate-13-acetate (PMA) at 100 ng/mL (HY-18739; MedChemExpress, Monmouth Junction, NJ) for 24 hours (h) (Park et al., 2023). Routine testing for mycoplasma was conducted on all the cell lines.

Cell viability assay

The CCK-8 test was used to assess the effect of plasma from TAO-A patients on the survival of MAFs. A total of 2 × 103 MAFs following the co-culture with the plasma were seeded for culture in independent triplicates for the culture of 0, 12, 24, 36 and 48 h, followed by the addition of 10-μ LCCK-8 working solution (CK04; Donjido, Kumamoto, Japan) and incubation for 3 h at 37 °C. The absorbance in each well was read at 450 nm using a microplate reader (iMark, Bio-Lad Laboratories, Inc., Hercules, CA, US).

Cell migration assay

Transwell assay was used to assess the migratory ability of plasma to MAFs in TAO-A patients. Cell migration assay was carried out utilizing a transwell chamber (8-μm pore size; EMD Millipore, Darmstadt, Germany). For migration assay, 3 × 104 MAFs in serum-free culture medium (200 μL) were seeded into the upper chamber, while the complete medium containing 10% FBS (700 μL) was added into the lower chamber. Subsequently, the chamber was incubated at 37 °C for 48 h, and the migrated cells were fixed by 4% paraformaldehyde (P1110; Solarbio Lifesciences, Beijing, China) at ambient temperature for 15 min. 0.1% crystal violet staining solution (G1063; Solarbio Lifesciences, Beijing, China) was applied for the staining of these cells at room temperature for 30 min, and the cells were visualized and counted under an inverted light microscope (Eclipse TS100; Nikon Corporation, Tokyo, Japan).

Cell immunofluorescence assay

MAFs or induced macrophages were seeded into the 24-well plates and cultured at 37 °C for 24 h. Hereafter, the cells were fixed in 4% paraformaldehyde for 10 min, blocked in 1% bovine serum albumin (IA0910; Solarbio Lifesciences, Beijing, China) for 30 min, and incubated with fluorescence-labeled primary antibodies at 4 °C overnight and then with secondary antibody at ambient temperature for 30 min in the dark. Next, glycerol (S2150; Solarbio Lifesciences, Beijing, China) was used to mount the cells, which were observed under a Zeiss LSM 510 fluorescence microscope (Carl Zeiss, Jena, Germany). The information of antibodies applied was listed in Table 1.

Table 1 Information of antibodies.

Antibody	Host species	Label	Dilution ratio	Catalog number and manufacturer	Application	
Anti-Ki67 antibody	Rabbit	Alexa Fluor® 488	1:100	ab197234, Abcam	Cell immunofluorescence assay	
Anti-Vimentin antibody	Rabbit	Alexa Fluor® 647	1:100	ab194719, Abcam	Cell immunofluorescence assay	
Anti-CD86 antibody	Rabbit	Alexa Fluor® 488	1:50	ab290990, Abcam	Cell immunofluorescence assay	
Anti-CD206 antibody	Rabbit	Alexa Fluor® 488	1:200	#36508, Cell Signaling Technology	Cell immunofluorescence assay	
Anti-CD11B antibody	Rabbit	Alexa Fluor® 647	1:200	NB110-89474AF647, Novus Biologicals	Cell immunofluorescence assay	
Goat anti-rabbit IgG H&L	Goat	/	1:2,000	ab6702, Abcam	Cell immunofluorescence assay	
Anti-PI3K antibody	Rabbit	/	1:1,000	#4292, Cell Signaling Technology	Western blotting	
Anti-p-PI3K antibody	Rabbit	/	1:1,000	ab182651, Abcam	Western blotting	
Anti-p-AKT antibody	Rabbit	/	1:2,000	#4060, Cell Signaling Technology	Western blotting	
Anti-AKT antibody	Rabbit	/	1:1,000	#4685, Cell Signaling Technology	Western blotting	
Anti-BAX antibody	Rabbit	/	1:10,000	ab32503, Abcam	Western blotting	
Anti-p-STAT1 antibody	Rabbit	/	1:10,000	ab109461, Abcam	Western blotting	
Anti-STAT1 antibody	Rabbit	/	1:10,000	ab109320, Abcam	Western blotting	
Anti-P65 antibody	Rabbit	/	1:10,000	ab32536, Abcam	Western blotting	
Anti-p-P65 antibody	Rabbit	/	1:1,000	ab76302, Abcam	Western blotting	
Anti-ERK1/2 antibody	Rabbit	/	1:10,000	ab184699, Abcam	Western blotting	
Anti-p-ERK1/2	Rabbit	/	1:2,000	#4370, Cell Signaling Technology	Western blotting	
Anti-GAPDH antibody	Rabbit	/	1:5,000	ab181602, Abcam	Western blotting	
Goat anti-rabbit IgG H&L	Goat	HRP	1:2,000	ab205718, Abcam	Western blotting	

ELISA

The contents of inflammatory cytokines including interleukin (IL)-1β (PI305), IL-6 (PI330), TNF-α (PT518), TGF-β (PT880) and IL-18 (PI558) within the plasma sample were quantified using the relevant ELISA assay kit purchased from Beyotime Institute (Shanghai, China) following the manuals.

Western blotting

Cells were lysed in RIPA lysis buffer (R0010; Solarbio Lifesciences, Beijing, China) and the protein concentrations were tested using BCA protein assay kit (PC0020; Solarbio Lifesciences, Beijing, China). Next, cell lysates were resolved via SDS-PAGE separation gel and transferred onto polyvinylidene difluoride (PVDF) membranes (YA1701; Solarbio Lifesciences, Beijing, China), which were blocked with 5% non-fat milk and probed with the primary antibodies at required dilution ratio at the temperature of 4 °C overnight. Horseradish peroxidase-conjugated secondary antibody was applied for amplifying the signals. Immunodetection was achieved using an ECL visualization agent (PE0010; Solarbio Lifesciences, Beijing, China) and ChemiDoc Imaging Systems (Bio-Rad Laboratories, Inc., Hercules, CA, USA). Finally, the densitometry on the protein bands was analyzed using ImageJ software 5.0 (Bio-Rad Laboratories, Inc., Hercules, CA, USA). See Table 1 for the information of antibodies.

RNA separation, cDNA synthesis and RT-qPCR

Rneasy Mini Kit (74104; Qiagen, Hilden, Germany) was employed to isolate total RNA from cells and total RNA (1 μg) was applied for the synthesis of cDNA utilizing the iScript™ cDNA Synthesis Kit (1708890; Bio-Rad Laboratories, Inc., Hercules, CA, USA) RT-qPCR was then carried out using iQ SYBR Green Master Mix (1708887; Bio-Rad Laboratories, Inc., Hercules, CA, USA) in CFX384 Touch Real-Time PCR System (Bio-Rad Laboratories, Inc., Hercules, CA, USA) at the following parameters for thermal cycling: at 95 °C for 3 min, and 40 repeated cycles at 95 °C for 10 seconds (s) and at 60 °C for 30 s. 2−ΔΔCt method was applied for calculating the relative mRNA levels, with GAPDH as a normalization control (Livak & Schmittgen, 2001; Amuthalakshmi, Sindhuja & Nalini, 2022). All primer sequences are shown in Table 2.

Table 2 Sequences of primers in qPCR assay.

Gene name/NCBI ID	Primers	
IL1B (3553)		
Forward	5′-CCACAGACCTTCCAGGAGAATG-3′	
Reverse	5′-GTGCAGTTCAGTGATCGTACAGG-3′	
IL6 (3569)		
Forward	5′-AGACAGCCACTCACCTCTTCAG-3′	
Reverse	5′-TTCTGCCAGTGCCTCTTTGCTG-3′	
IL18 (3606)		
Forward	5′-GATAGCCAGCCTAGAGGTATGG-3′	
Reverse	5′-CCTTGATGTTATCAGGAGGATTCA-3′	
TGFB (7040)		
Forward	5′-TACCTGAACCCGTGTTGCTCTC-3′	
Reverse	5′-GTTGCTGAGGTATCGCCAGGAA-3′	
MYBPC3 (4607)		
Forward	5′-AACCTGTCAGCCAAGCTCCACT-3′	
Reverse	5′-CCACAATGGTGTCTGGTATGCG-3′	
COL1A1 (1277)		
Forward	5′-GATTCCCTGGACCTAAAGGTGC-3′	
Reverse	5′-AGCCTCTCCATCTTTGCCAGCA-3′	
ACTA2 (59)		
Forward	5′-CTATGCCTCTGGACGCACAACT-3′	
Reverse	5′-CAGATCCAGACGCATGATGGCA-3′	
VEGFA (7422)		
Forward	5′-TTGCCTTGCTGCTCTACCTCCA-3′	
Reverse	5′-GATGGCAGTAGCTGCGCTGATA-3′	
GAPDH (2597)		
Forward	5′-GTCTCCTCTGACTTCAACAGCG -3′	
Reverse	5′-ACCACCCTGTTGCTGTAGCCAA-3′	

Statistical analysis

All the experiments were performed in independent triplicates and all the data were represented as mean ± standard deviation. Two-group differences were compared with unpaired student’s t tests. GraphPad Prism 6.0 software (GraphPad Software, Inc., La Jolla, CA, USA) was applied for statistical analysis. A p-value lower than 0.05 indicated a statistically significant difference.

Results

Characterization on the plasma

Detailed procedures for plasma characterization were presented in Fig. 1A. ELISA was applied to calculate the levels of pro-inflammatory cytokines including IL-1β, IL-6, TNF-α, TGF-β and IL-18. Notably, the levels of these cytokines were relatively higher in the plasma from TAO-A patients as compared to those in HC (Fig. 1B, p < 0.05), indicating that the plasma from TAO-A patients had stronger pro-inflammatory properties.

Figure 1 Characterization on the plasma.

(A) The detailed procedures for plasma characterization. (B) The assay of ELISA was applied to calculate the levels of pro-inflammatory cytokines including IL-1β, IL-6, TNF-α, TGF-β and IL-18 in TAO-A patients and healthy control (n = 5). Data are mean ± standard deviation from three independent experiments. *p < 0.05.

Effects of the plasma derived from TAO-A patients on the survival and migration of MAFs

The procedures for co-culturing MAFs with the plasmas of the two groups of patients were illustrated in Fig. 2A. Next, the effects of these plasmas on the survival and migration of MAFs were explored. The relevant results from CCK-8 and Transwell assays demonstrated that the plasma of TAO-A patients could evidently promote the viability and migration of MAFs (Figs. 2B–2D, p < 0.05). Ki67 is a cell proliferation marker commonly used to mark proliferative cells (Li et al., 2015). VIM is an intermediate fiber protein widely expressed in fibroblasts and other mesenchymal cells (Ostrowska-Podhorodecka & McCulloch, 2021). Here, Ki67+VIM+ MAFs were the MAFs that were proliferative (Ki67-positive) and had fibroblast properties (VIM-positive). These double-positive cells represented a highly active class of MAFs involved in the structural support of the TME and accelerated the expansion of the TME and tumor progression through cell proliferation. Further, the results from immunofluorescence assay confirmed that the co-culture of the plasma from TAO-A patients increased the number of Ki67+VIM+ MAFs (Figs. 2E and 2F, p < 0.01).

Figure 2 Effects of TAO-A patients-derived plasma on the survival and migration of MAFs.

(A) The detailed procedures for plasma co-culture with MAFs. (B) Relative viability of MAFs under different co-culture schemes at 12, 24 and 36 h. (C and D) Number of migrated MAFs under different co-culture schemes at 48 h. (E and F) Number of Ki67+VIM+ MAFs under different co-culture schemes based on immunofluorescence assay. The scale is 50 μm. Data are mean ± standard deviation from three independent experiments. *p < 0.05, **p < 0.01, ***p < 0.001.

Effects of the plasma from TAO-A patients on proteins related to PI3K/AKT pathway in MAFs and apoptosis

Subsequently, we analyzed the relevant pathway and molecular mechanisms underlying the effects of the plasma from TAO-A patients on MAFs. PI3K/AKT pathway is involved in the regulation of cell survival and migration (Ma et al., 2021). Here, we focused on PI3K/AKT pathway and the phosphorylation of relevant proteins PI3K and AKT were measured accordingly. It was found that the plasma of TAO-A patients visibly enhanced the phosphorylation of the two proteins in MAFs (Figs. 3A and 3B, p < 0.01). Bax, an apoptosis-related protein, was additionally used to determine the effects of the plasma from TAO-A patients on MAFs, and the results showed that Bax was reduced in MAFs following the culture with the plasma derived from TAO-A patients (Figs. 3A and 3B, p < 0.01). These results suggested that the plasma from TAO-A patients inhibited the apoptosis of MAFs through PI3K/AKT signaling pathway.

Figure 3 Effects of TAO-A patients-derived plasma on proteins related to PI3K/AKT pathway and apoptosis in MAFs.

(A and B) Effects of TAO-A patients-derived plasma on proteins related to PI3K/AKT pathway and apoptosis in different groups of MAFs were explored based on the results of western blotting assay. Data are mean ± standard deviation from three independent experiments. **p < 0.01.

Effects of the plasma derived from TAO-A patients on inflammation-related and potential downstream genes in MAFs

Then we measured the levels of genes related to inflammation in MAFs co-cultured with different types of plasma using qPCR, and observed that the co-culture with the plasma of TAO-A patients upregulated the expression of inflammatory cytokines including IL1B, IL6, IL18 and TGFB (Fig. 4A, p < 0.05) and that of the potential downstream genes MYBPC3, COL1A1, ACTA2 and VEGFA (Fig. 4B, p < 0.05). This indicated that the plasma from TAO-A patients enhanced the pro-inflammatory response of MAFs and their role in the TME.

Figure 4 Effects of TAO-A patients-derived plasma on inflammation-related and potential downstream genes in MAFs.

(A) Levels of pro-inflammatory cytokines in different groups of MAFs were calculated based on qPCR. (B) Levels of potential downstream genes in different groups of MAFs were calculated based on qPCR. Data are mean ± standard deviation from three independent experiments. *p < 0.05, **p < 0.01.

Effects of the plasma of TAO-A patients on M1 macrophage polarization

Additionally, the macrophage polarization status following the co-culture with different types of plasma was analyzed, and the co-culture procedures were shown in Fig. 5A. The macrophage polarization status was reflected by the mean fluorescence intensity (MFI) on CD86 (M1 polarization marker) and CD206 (M2 polarization indicator) using immunofluorescence. According to the results of immunofluorescence assay, the plasma derived from TAO-A patients evidently enhanced the MFI of CD86 but reduced that of CD206 (Figs. 5B–5E, p < 0.05).

Figure 5 Effects of TAO-A patients-derived plasma on M1 macrophage polarization.

(A) Co-culture schemes of induced macrophages with different groups of plasma. (B and C) CD86 MFI in macrophages co-cultured with different groups of plasma. (D and E) CD206 MFI in macrophages co-cultured with different groups of plasma. The scale is 50 μm. Data are mean ± standard deviation from three independent experiments. *p < 0.05.

Effects of the plasma derived from TAO-A patients on possible downstream pathways in macrophages

We explored the relevant proteins in macrophages following the co-culture, with a focus on STAT, p65 and ERK pathways. The activity of these signaling pathways was measured to assess whether the plasma from TAO-A patients enhanced the pro-carcinogenic properties of MAFs through the activation of these pathways to alter the TME and accelerate cancer progression. Based on Western blotting assay, it was found that the co-culture of the plasma from TAO-A patients enhanced the phosphorylation of STAT1, P65 and ERK1/2 (Figs. 6A and 6B, p < 0.01).

Figure 6 Effects of TAO-A-patients-derived plasma on possible downstream pathways in macrophages.

(A and B) Relevant proteins to STAT, p65 and ERK pathways in macrophages following the co-culture were quantified based on western blotting assay. Data are mean ± standard deviation from three independent experiments. **p < 0.01.

Discussion

TAO is a prevalent autoimmune inflammatory disorders in the orbit and a leading cause of orbital and strabismus symptoms to adults (Kyriakos et al., 2022; Hennein & Robbins, 2022). It has been noted that plasma exosomes from TAO-A patients could trigger inflammation in orbital fibroblasts (Wei et al., 2024). Based on previous findings, we speculated that the plasma from TAO-A patients could also initiate inflammation to promote the development of melanoma. In the current study, the effects of plasma from TAO-A patients on the survival of MAFs were analyzed, and it was observed that the plasma of TAO-A patients promoted the survival and migration of MAFs and enhanced M1 macrophage polarization, thereby accelerating melanoma progression in vitro.

Melanoma is one of the most aggressive and therapy-resistant cancers that has no adequate treatment available (Shannan et al., 2016; Grossi et al., 2023). The dynamic relationship between melanoma and fibroblast has the potential to improve the therapeutic options for patients with the cancer (Papaccio et al., 2021). The current study used isolated MAFs to investigate their specific effects on melanoma progression. MAFs are involved in tumor-promoting inflammation and anti-tumor immunity (Zhou et al., 2024). Specifically, MAFs can reduce tumor cell susceptibility to natural killer cell-mediated killing via secreting matrix metalloproteinases (Ziani et al., 2017). Meanwhile, MAFs could also impair the function of CD8+ T cells and modify the levels of immune checkpoint regulators by increasing arginase activity (Érsek et al., 2021). Additionally, mesenchymal-stromal cell-like MAFs increase IL-10 production by macrophages dependent on cyclooxygenase/indoleamine 2,3-dioxygenase (Çakır et al., 2021). As for the modulator of MAFs, it has been found that HSP90/IKK-rich small extracellular vesicles can activate those pro-angiogenic MAFs via the NF-κB/CXCL1 axis (Tang et al., 2022). In the current study, the co-culture of MAFs with the plasma from TAO-A patients evidently promoted the survival and migration of MAFs, as we observed that the cell viability and the numbers of migrated cells and Ki67+VIM+ MAFs were increased. Further exploration on the relevant pathways showed that the phosphorylation levels of PI3K and AKT in MAFs were elevated, indicating the potential involvement of PI3K/AKT pathway in the initiation and therapeutic resistance of melanoma (Davies, 2012).

Additionally, melanoma cells can interact with and are dependent on seemingly normal cells in their TME, thereby permitting the acquisition of their microenvironment (Brandner & Haass, 2013). In the context of skin cancers, there are some common characteristics of TME such as the presence of tumor-associated macrophages (Georgescu et al., 2020). Macrophages not only triggers the adaptive immune response and enhances the killing of tumor cells, but also promotes tumorigenesis and metastasis of melanoma when affected by the factors existing in the TME of melanoma (Habib et al., 2024). Further, mounting evidence indicated that macrophages can be classified into two distinctly different types (M1 and M2) (Wang et al., 2017). In the current study, elevated MFI of CD86 suggested M1 polarization in macrophages following the co-culture with the plasma from TAO-A patients, which was consistent with some existing studies describing the role of M1 macrophage in driving protumor inflammation of melanoma cells via TNFR-NF-κB signaling (Kainulainen et al., 2022). Further examination on the relevant pathways involved showed elevated levels of phosphorylation of STAT1, p65 and ERK1/2, which all participate in macrophage M1 polarization under diverse circumstances (Kainulainen et al., 2022; Huangfu et al., 2020; Lu et al., 2023).

Conclusion

In conclusion, we demonstrated that the plasma from TAO-A patients profoundly influenced the survival and inflammation in MAFs. However, some limitations still existed in this study. For instance, this study only explored the effects of the plasma from TAO-A patients on the survival and inflammation of MAFs, but the specific molecules contributing to such effects were not investigated. Also, the study was conducted based on in vitro assays, therefore further in vivo experiments should be incorporated to validate the current findings.

Supplemental Information

Supplemental Information 1 MIQE checklist.

Abbreviations

TME tumor microenvironment

CAFs cancer-associated fibroblasts

MAFs Melanoma-associated fibroblasts

TAO-A active thyroid-associated orbitopathy

HC healthy control

EDTA ethylenediaminetetraacetic acid

PBS phosphate-buffered saline

CCK-8 cell counting kit-8

VIM Vimentin

IL interleukin

TNF tumor necrosis factor

TGF transforming growth factor

MFI mean fluorescence intensity

Additional Information and Declarations

Competing Interests

Author Contributions

Human Ethics

Data Availability

The authors declare that they have no competing interests.

Huifang Chen conceived and designed the experiments, performed the experiments, analyzed the data, prepared figures and/or tables, authored or reviewed drafts of the article, and approved the final draft.

Shiyuan Chen conceived and designed the experiments, analyzed the data, prepared figures and/or tables, authored or reviewed drafts of the article, and approved the final draft.

Zhenfeng Liu performed the experiments, analyzed the data, authored or reviewed drafts of the article, and approved the final draft.

The following information was supplied relating to ethical approvals (i.e., approving body and any reference numbers):

The study was supported by Guangxi University of Science and Technology Medical Ethics Committee (No. YX20230301H015).

The following information was supplied regarding data availability:

The raw data is available at GitHub, Zenodo, and figshare:

- https://github.com/1zf-gif/Raw-data.git

- 1zf-gif. (2024). 1zf-gif/Raw-data: Raw data (v.1.1.0). Zenodo. https://doi.org/10.5281/zenodo.13324703

- Chen, Huifang; Chen, Shiyuan; Liu, Zhenfeng (2024). Raw experimental data. figshare. Figure. https://doi.org/10.6084/m9.figshare.26793838.v1.

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
