# Peer review of "The effects of plasma from patients with active thyroid-associated orbitopathy on the survival and inflammation of melanoma-associated fibroblasts"

_PeerJ, doi:10.7717/peerj.18612_

## Round 0.1 · original submission · Major Revisions

After careful consideration of the three reviewers' comments, I have decided that your manuscript requires major revisions before it can be considered for publication. Please thoroughly address all reviewers' concerns, particularly those suggesting significant changes. Submit a revised manuscript along with a point-by-point response detailing how you've addressed each comment.

Reviewer 1 ·

Basic reporting

In this study, the author demonstrated that the plasma of patients with active thyroid-associated orbitopathy can enhance the survival and migration, and inflammatory of melanoma-associated fibroblasts for tumor progression. The experiment is reasonable and the arguments are sound. However, the author still needs to refine the manuscript before publication.
1. Line 29-34, Please describe clearly the method used in each experiment, such as inflammatory cytokines were detected by ELISA.
2. Line 47-55, the author indicated the tumor microenvironment (TME) is crucial for tumor growth and therapy. So, the risk factors, origin and TME feature of Melanoma is what. In addition, the introduction about TME is not strongly related to the purpose of this article. Introduction logic confusion suggest rewriting this section.
3. The title “Involvement of plasma from patients with active thyroid-associated orbitopathy in survival and inflammation of melanoma-associated fibroblasts” revealed the plasma of patients with thyroid-associated orbitopathy affected melanoma-associated fibroblasts. But the author does not make clear the relationship between thyroid-associated orbitopathy and melanoma in introduction. The purpose of this study is very vague.

Experimental design

4. The characteristic of fibroblasts and melanoma-associated fibroblasts is what respectively. The inflammation characteristic of melanoma-associated fibroblasts is what. Werther it can increase the exhaustion of immune killer cells.
5. Line 64-66, What are the specific mechanisms of melanoma-associated fibroblasts (MAFs) promoting immune evasion and proliferation of tumor cells. Is this paper trying to verify related mechanisms or discover new ones of these mechanisms. Line 73, Please add the experimental procedure, purpose and some important results at the end of this paragraph.
6. Line 64, What are the cancer-promoting features of melanoma-associated fibroblasts. Whether the function of promoting cancer is achieved by affecting immune infiltration or critical path activation.
7. Line 82-162, Please briefly describe the purpose of each part of the experiment, such as the ELISA was used for the plasma characterization. The Human monocyte THP-1 was induced into macrophages, what role does it play in this article.
8. Line 181, the distinction of MAFs and the Ki67+VIM+ MAFs, please give a proper explanation. Line 187, the PI3K/AKT pathway regulated the cell survival and migration, please add this sentence to complete the logic of this article.
9. Line 211, the role of STAT1, P65 and ERK1/2 pathway is what. How do these pathways affect cancer progression.

Validity of the findings

no comment

Additional comments

10. What is the significance of this article. How will it guide future research.

Reviewer 2 ·

Basic reporting

The purpose of this study was to explore the effects of plasma from patients with active thyroid-associated orbitopathy (TAO-A) on the survival and inflammation of melanoma-associated fibroblasts (MAFs). This study revealed that TAO-A patients-derived plasmas were characterized by elevated pro-inflammatory cytokines levels and promoted the survival and aggravated the inflammation in MAFs, concurrent with the enhanced levels of PI3K and AKT yet the diminished expression of Bax. In addition, co-culture of these plasma evidently promoted macrophage M1 polarization, along with the increased phosphorylation degrees of STAT1, P65 and ERK1/2. This study could be the cornerstone for some future studies delving into the relevant mechanisms. In my point of view, this study has certain innovation and scientific research value, and the experimental design is rigorous and logical.
1. The abstract section of this study is too laconic, and a more specific description is suggested to supplement. For instance, what methods are used to determine and evaluate the levels of pro-inflammation cytokines, the cell viability, cell migration, and the levels of relevant pathway-related proteins?
2. This study explores the effects of plasma from patients with TAO-A on the survival and inflammation of MAFs, but the relevant introduction about TAO-A is deficient. It is recommended to add some detailed background information about TAO-A into the first paragraph of the introductory section, such as the pathophysiological mechanism, incidence rate, and current treatment challenges. Meanwhile, the “active thyroid-associated orbitopathy” also is suggested to serve as one Keyword of this study.
3. In the introductory section, before introduce the results of this study, it is suggested that a brief statement of the specific experimental method process is needed.
4. In the Material and methods section, the relevant literatures for each assay are recommended to cite for further reference by the reader.

Experimental design

5. A cell concentration of 2×103 MAFs is used to test the cell viability, while a concentration of 3×104 MAFs is utilized to assess the cell migration. Could the author explain why these two different concentrations are chosen for the experiments?
6. The description of results section is relatively straightforward; accordingly, it is recommended to describe the results in detail and supplement a short summary at the end of each result section to make the article more coherent.

Validity of the findings

7. In lines 180-181, the results from immunofluorescence assay have confirmed that the co-culture of TAO-A patients-derived plasma led to the increased number of Ki67+VIM+ MAFs. What are the implications of the enhancement of Ki67+VIM+ MAFs? And, what is the connection between the enhancement of Ki67+VIM+ MAFs and cell migration? Please add some discussions to the manuscript.
8. In the Figure 2E and Figure 5C, E, the scale bars of fluorescence images are required to supplement into the Figure captions.
9. Why measure the expression levels of the potential downstream genes (MYBPC3, COL1A1, ACTA2, VEGFA), whether these genes have impacts in melanoma progression? Please add appropriate discussion.

Reviewer 3 ·

Basic reporting

The authors present their experimental data obtained in simplistic culture models. The critical part is - as far as I can understand - the employment of a complex biological mixture of proinflammatory molecules from thyroid-associated orbitopathy patients´ plasma. The characterisation of the content is rather unsatisfactory with five principal, mostly well-known, inflammatory mediators (IL-1, IL-6, IL18, TGFb, TNFa). It is neither specific for orbitopathy nor novel and previously unknown in advanced melanoma patients.
The manuscript is acknowledges reasonably well the previous relevant research conducted by others.
The article is not very easy to read, it requires extensive language editing. The graphical part is standard, I believe there are more eye-catching graph plotting applications available even for free. Moreover, panel A in Figures 1,2,5 (the experiment's time course) seems rather unnecessary.
In general, the reasoning behind such a study design and general hypothesis is not very clear.

Experimental design

The originality of the manuscript is relatively low; the methods are routine/basic. I believe it does not sufficiently meet the standards for PerJ publication.
The proinflammatory mediators and their impact on melanoma (including stromal components) are extensively studied earlier by others. The principal research question of this manuscript remains somewhat elusive. I greatly miss some links of this research to the clinical aspects of melanoma.
The method description seems to be mostly sufficient. The HC (healthy control) selection strategy is not mentioned. The cell culture comments only on THP1 macrophages. I believe that it is necessary to use at least one other macrophage cell line (or preferably PBMC) for result verification. It is not explicitly stated that all primary cells were also routinely tested for Mycoplasma contamination.
Regarding ethical standards, I do not raise any objections.

Validity of the findings

The authors have demonstrated that TAO serum contains higher levels of proinflammatory mediators. The mediators can increase (at least temporarily) fibroblast proliferation. The proinflammatory mediators can also affect macrophage polarization; however, I believe CD206 is not a completely informative marker in the given context. Moreover, this M2-polarization marker was increased in HC - it is somewhat surprising, and the authors do not discuss this finding sufficiently.
In conclusion, the presented evidence is weak and does not form a sufficient base for scientifically sound conclusions.

---

## Round 0.2 · accepted · Accept

After careful consideration of your revised manuscript and responses to reviewers' comments, I am pleased to inform you that your manuscript has been accepted for publication. While Reviewer 3 raised valid points about methodology simplicity and novelty in their initial review, you have clearly articulated the innovative aspects of your work - specifically the first exploration of TAO-A plasma effects on melanoma microenvironment - and acknowledged current limitations while outlining plans for future validation studies. The thorough responses you provided to all concerns, particularly regarding experimental design rationale and methodology choices, were appropriate and comprehensive. Combined with the positive recommendations from two other reviewers who endorsed your revisions, I am confident in recommending your manuscript for publication.

Reviewer 1 ·

Basic reporting

In this study, the authors demonstrated that plasma from patients with active thyroid associated orbital disease can increase the survival and migration rates of melanoma associated fibroblasts and promote inflammation in tumor progression. The experiment is reasonable, and the argument is reasonable. The revised manuscript is close to perfection and meets the publishing requirements.

Experimental design

no comment

Validity of the findings

no comment

Reviewer 2 ·

Basic reporting

The purpose of this study was to explore the effects of plasma from patients with active thyroid-associated orbitopathy (TAO-A) on the survival and inflammation of melanoma-associated fibroblasts (MAFs). This study revealed that TAO-A patients-derived plasmas were characterized by elevated pro-inflammatory cytokines levels and promoted the survival and aggravated the inflammation in MAFs, concurrent with the enhanced levels of PI3K and AKT yet the diminished expression of Bax. In addition, co-culture of these plasma evidently promoted macrophage M1 polarization, along with the increased phosphorylation degrees of STAT1, P65 and ERK1/2. This study could be the cornerstone for some future studies delving into the relevant mechanisms. In my point of view, this study has certain innovation and scientific research value, and the experimental design is rigorous and logical.The author has responded to all the reviewers' comments, and I believe they have resolved the vast majority of the issues.

Experimental design

no comment

Validity of the findings

no comment